# Ruthenium Decorated Polypyrrole Nanoparticles for Highly Sensitive Hydrogen Gas Sensors Using Component Ratio and Protonation Control

**DOI:** 10.3390/polym12061427

**Published:** 2020-06-26

**Authors:** Jungkyun Oh, Jun Seop Lee, Jyongsik Jang

**Affiliations:** 1School of Chemical and Biological Engineering, Seoul National University, 1 Gwanak-ro, Gwanak-gu, Seoul 151-742, Korea; freezeoh@snu.ac.kr; 2Department of Materials Science and Engineering, Gachon University, 1342 Seongnam-Daero, Sujeong-Gu, Seongnam-Si, Gyeonggi-Do 13120, Korea

**Keywords:** ruthenium, polypyrrole, protonation, hydrogen gas, chemical sensor, nanocomposite

## Abstract

Despite being highly flammable at lower concentrations and causing suffocation at higher concentrations, hydrogen gas continues to play an important role in various industrial processes. Therefore, an appropriate monitoring system is crucial for processes that use hydrogen. In this study, we found a nanocomposite comprising of ruthenium nanoclusters decorated on carboxyl polypyrrole nanoparticles (Ru_CPPy) to be successful in detecting hydrogen gas through a simple sonochemistry method. We found that the morphology and density control of the ruthenium component increased the active surface area to the target analyte (hydrogen molecule). Carboxyl polypyrrole (CPPy) in the nanocomposite was protonated to increase the charge transfer rate during gas detection. This material-based sensor electrode was highly sensitive (down to 0.5 ppm) toward hydrogen gas and had a fast response and recovery time under ambient conditions. The sensing ability of the electrode was maintained up to 15 days without structure deformations.

## 1. Introduction

Hydrogen is one of the most popular raw materials used in diverse industrial processes due to its important advantages, such as its high availability in nature, high conversion efficiency, non-toxicity, and feasibility of development from renewable sources [1,2]. Recently, the application of hydrogen gas in hydrogen-based fuel cell vehicles has received increasing attention due to its high conversion efficiency without the ramifications of hazardous gases. However, hydrogen is very dangerous, as it demonstrates strong flammability and explosiveness, and even causes asphyxiation at high concentrations [3,4]. Therefore, rapid detection of hydrogen molecules at ambient conditions is of critical importance to monitor concentration changes in real-time.

Hydrogen gas sensors that use various sensing materials have been widely studied by many research groups for over two decades [5,6,7,8]. Recently, palladium nanomaterial-based sensors that can detect down to ppm level at room temperature have been developed [9,10,11]. Cho et al. synthesized half-pipe palladium nanotube-based networks for application into a flexible hydrogen gas sensor [11]. Despite the high sensitivity of existing sensors to the target analyte, some important technical issues still need to be addressed by further research and development [12,13,14,15,16]. For example, a drop in sensing signals during repetitive detection of hydrogen causes a delayed response in the sensing device [17]. The change in structure of the sensing materials during exposure to hydrogen also generates a variation in response time, as well as simultaneous delays in the sensing signal [18]. Therefore, to achieve advanced properties and performance, innovative materials are necessary for hydrogen sensor devices. In this respect, composite nanomaterials containing both organic and inorganic materials have been constantly studied and reported due to their excellent functionalities originating from each component [19,20]. As a sensor electrode material, the organic component (i.e., sp^2^ carbon) acts as a charge transfer pathway and a template, whereas the inorganic component operates as binding site to target analyte. A new synthetic process of sensor electrode materials has recently been developed using a facile sonochemical method [21,22,23].

Conductive polymer nanomaterials can be used for the organic component as a template for nanocomposites and a charge transfer pathway due to the conjugated polymer backbone structure in the sensor transducer [24,25,26,27]. Among them, polypyrrole is one of the most well-known conducting polymers that comprises five-membered heterocyclic rings and exhibits beneficial characteristics in a hydrogen sensing system [28,29,30,31]. The change in the doping level of the polymer chain can modify the number of generated signals during the detection of hydrogen molecules and can thus improve the stability of the sensor device [32,33]. Ruthenium is often used as an inorganic component, which is a transition metal included in the platinum group of the periodic table. It is inert to most other chemicals and has emerged as a substitute for gold and platinum, as it is more cost-effective than noble metals [34,35]. Ruthenium has been recently studied as a hydrogen gas sensing material in sensor devices due to having similar properties as platinum in the catalytic reaction with hydrogen molecules [36].

In our research, we identified a method for the facile synthesis of a highly sensitive hydrogen gas sensor electrode material composed of ruthenium nanoclusters decorated with carboxyl polypyrrole nanoparticles (Ru_CPPy). The size and number of ruthenium nanoclusters were optimized using sonochemical reduction by controlling the amount of ruthenium precursors. The chemical state of the particles was changed through the solution-based doping method to confirm the effect of the proton amount on the charge transfer rate of the sensor electrode. The protonated Ru_CPPy-based electrode showed a high sensitivity toward hydrogen gas, a low detection limit (down to 0.5 ppm), and fast response and recovery times. Additionally, the performance of the sensor electrode presented a similar value after repeated exposure to hydrogen gas (up to 15 days) due to its uniformly decorated Ru nanoclusters on the carboxyl polypyrrole surface.

## 2. Materials and Methods

### 2.1. Materials

For this study, we used FeCl_3_ (97%), poly(vinyl alcohol) (PVA, Mw 9000), dodecyl trimethylammonium bromide (DTAB, ≥ 98%), pyrrole (98%), and NaBH_4_ (≥ 98%) from Aldrich Chemical Company (St. Louis, MI, USA) and used each as purchased. Ruthenium (III) chloride hydrate (99.98%) was acquired from Aldrich Chemical Company, as well. Ammonium persulfate (APS, 98%) and pyrrole-3-carboxylic acid were obtained from Sigma-Aldrich and Acros Organics (Geel, Belgium). Sodium hydroxide (NaOH, 95%) and hydrochloric acid (HCl, 35~37%) were purchased from Samchun Pure Chemical Company (Seoul, Korea).

### 2.2. Synthesis of Ru_CPPy

The carboxyl polypyrrole nanoparticles (CPPyNP) that had a 65 nm diameter were synthesized with PVA, FeCl_3_, and a mixture of pyrrole and pyrrole-3-carboxylic acid monomers using a method described in a previous study [37]. A total of 3 mg of the CPPyNP was added to aqueous solutions of RuCl_3_ with different concentrations. Then, 1 mg of NaBH_4_ was injected into the mixed solution via ultrasonication for 30 min. Finally, the mixed solution was washed three times using a centrifuge (10,000 rpm, 20 min) with water then ethanol. It was then dried for 12 h at 60 °C.

### 2.3. Protonation Control of Ru_CPPy

Ru_CPPy was added to different pH aqueous solvents and was sonicated for 1 h to disperse the nanoparticles uniformly in the solvent with the exception at pH 7 of HCl and NaOH, as these were used for realizing acidic and basic conditions, respectively. For the pH 7 solution, we used commercial phosphate buffers (Samchun Co., Seoul, Korea). The solutions with Ru_CPPy were then stirred steadily at a low rate for another 1 h to enable a complete reaction between the particles and solvent. The mixed solution was then put through the centrifuge and was dried at 60 °C in an oven overnight.

### 2.4. Electrical Sensing Measurements of the Ru_CPPy-Based Electrode

To obtain a uniformly coated sensor electrode array, a spin-coating procedure was conducted. Then, 10 µL of a 0.1 wt % Ru_CPPy solution was deposited onto the electrode and a subsequent spin-coating was conducted (2000 rpm, 60 s). The electrode was then dried at 60 °C in an inert atmosphere for 6 h to obtain good electrical ohmic contact between the Ru_CPPy and electrodes. To confirm the influence of hydrogen gas exposure on the electrical properties, the sensor electrodes were placed in a vacuum chamber that was designed specifically for gas sensing with a vapor inlet/outlet pressure of 10^0^ Torr. The concentrations of hydrogen gas (in the range of 0.5–100 ppm) were controlled through a mass flow controller system (MFC, KNH Instruments, Pocheon, Korea). The target gas was injected and the gas inlet was opened until saturation was reached (observed as constant resistance). The system was then purged with compressed mixing gas (N_2_:O_2_ = 9:1) for the same time duration.

### 2.5. Characterization

Field-emission scanning electron microscopy (FE-SEM) images were collected using a JEOL JEM-200CX high performance transmission electron microscope (JEOL Ltd., Tokyo, Japan). Transmission electron microscopy (TEM) images were acquired by the JEM-2100 (JEOL) installed at the National Center for Inter-university Research Facilities at Seoul National University. The X-ray photoelectron spectra (XPS) were recorded using a M16XHF-SRA (MAC Science Co., Yokohama, Japan). The X-ray diffraction (XRD) patterns were obtained using a M18XHF SRA (MAC Science Co.). The four-probe method was used to measure the electrical conductivity with a source meter (Keithly Instruments Inc., Cleveland, OH, USA). Then, a polymer film consisting of nanoparticles (ca. 120 nm thickness) was manufactured using spin-coating (2000 rpm, 60 s) of the mixed solution (2 mL) on a 1 × 1 cm area of glass substrate. The electrical conductivity of the thin film was then measured. Next, the Fourier transform infrared (FTIR) spectra were measured using a PerkinElmer Frontier spectrophotometer in attenuated total reflection mode. The Raman spectra were obtained using a LabRam Aramis (Horiba Jobin Yvon) spectrometer.

## 3. Results and Discussion

### 3.1. Fabrication of Ru_CPPy

Appendix A shows the schematic illustration of ruthenium nanoclusters decorated on the CPPyNP surface obtained through facile sonochemical reduction. The CPPyNPs that were prepared by the monodisperse method demonstrated a uniform diameter of ca. 65 nm without any wrinkles or cracks (Figure 1a and Appendix A). The aqueous solution containing CPPyNP mixed with different concentrations of the ruthenium precursor (RuCl_3_) was stirred at room temperature to generate a charge–charge interaction between the ruthenium ions (Ru^3+^) and the negative charge of the oxygen atom of the carboxyl group in the polymer chain [38]. The mixed solution was then sonicated with a minimal amount of a reductant (NaBH_4_) to reduce Ru^3+^ to ruthenium nanoclusters (Ru^0^). This was achieved using the cavitation bubbles with H^+^ and OH^−^ ions in the solution [39]. To determine the role of CPPy in Ru particle generation, the reaction was conducted excluding CPPyNP, and a large Ru particle (ca. 800 nm) was generated with a clump (Appendix A). This proved that CPPyNP is able to act as a framework for creating uniform Ru nanoparticles.

The size and density of the ruthenium nanoclusters were regulated by the amount of RuCl_3_ in the solution. The nanoparticles were denoted based on the amount of RuCl_3_ in the solution; the nanoparticles with 0.5 mg of RuCl_3_ are denoted as Ru_CPPy_0.5, 1.5 mg as Ru_CPPy_1.5, 3.0 mg as Ru_CPPy_3.0, 4.0 mg as Ru_CPPy_4.0, and 5.0 mg as Ru_CPPy_5.0. The size and density of the decorated nanoclusters consistently increased as the concentration of RuCl_3_ increased (Figure 1b–f). Particularly, the size of the nanoclusters was enhanced until it reached 3 mg of RuCl_3_ solution (from 2–6 nm) while maintaining a minimal deviation (less than 0.35 nm). However, the ruthenium nanoclusters generated from a large amount of RuCl_3_ (4.0 mg) demonstrated a large scale variation and self-aggregation, rather than decorations on the CPPyNP surface (Appendix A). The crystalline structure of the nanoparticles (CPPyNP and Ru_CPPy) is shown in Appendix A. Both nanoparticles showed broad peaks from approximately 25.6°, thereby indicating the amorphous structure of carboxyl polypyrrole. Contrastingly, the Ru_CPPy showed diffraction peaks that corresponded with the (100), (002), (101), and (102) planes of ruthenium nanoparticle crystals (JCPDS card no. 06–0663) and were indexed to the hexagonal close-packed (hcp) phase. This phenomenon indicated that the Ru nanoparticles (Ru^0^) were well formed on the CPPyNP surface. The HR-TEM results of the CPPyNP also demonstrated an inter-planar spacing of 0.20 nm for the (101) plane of hcp, confirming the development of pure crystalline Ru nanoclusters after the facile sonochemical synthesis (Appendix A).

### 3.2. Protonation Control of Ru_CPPy

The charge carrier density and the mobility of the carboxyl polypyrrole chain structure changed reversibly with different proton amounts. The Ru_CPPy was dispersed in aqueous solvents with diverse pH values to transform the chemical state and morphology of the nanoparticles. Figure 2a–c show the high-resolution N 1s XPS of the particles, thus confirming the chemical state change of the carboxyl polypyrrole chain with different pH values. Two noticeable nitrogen peaks were observed in the deconvolution of the N 1s peak linked with the pyrrolic nitrogen (–NH–) in the pyrrole ring at 400.3 eV and imine nitrogen (–N=) at 398.0 eV. With these data, the proton amounts in the nanoparticles could be calculated in terms of the area ratio of the imine peak over the total nitrogen peaks (–N=/N_total_) [40]. The ratio of imine over total nitrogen (–N=/N_total_) corresponds to the proton amount of carboxyl polypyrrole, as the –N= structure in the polymer chain decreases due to positive charges in the pyrrole rings. Thus, a lower value of –N=/N_total_ implies a higher proton amount in the polymer chain. When treated with a base, the value of –N=/N_total_ was found to be higher than those under other conditions because positive nitrogen (–N^+^–) in the pyrrole structure decreases (i.e., deprotonation) at high pH levels. Contrastingly, when treated with an acid that causes a lower value of –N=/N_total_ (i.e., protonation), the positive nitrogen increased. Moreover, the Raman spectra of the particles also showed a change in the chemical state of carboxyl polypyrrole chains at different pH values (Appendix Aa). The peak at 1555 cm^−1^ that corresponds to the C=C stretching vibrations increased at lower pH solutions, as the amount of C=C stretching in the polymer chain increased with more protons in the polymer chain. The FTIR also demonstrated an enhancement and shift in C=C stretching vibrations at 1555 cm^−1^ during protonation treatment (Appendix Ab) [41]. Simultaneously, the chemical state of the Ru component maintained a constant value at different pH solutions. The high resolution XPS spectra of the Ru 3d and Ru 3p peaks also suggested that the valance state of ruthenium (Ru^0^) remained constant during the chemical treatment (Figure 2d–e).

To confirm the morphology transformations of Ru_CPPy with varying pH values, TEM and HR-TEM measurements were recorded for each nanoparticle (Figure 3). The morphology of the nanoparticles also suggested that the CPPyNP and ruthenium nanoclusters with diameters of 65 nm and 6 nm, respectively, were preserved at different pH values. Specifically, HR-TEM images of the ruthenium nanoclusters indicated an inter-planar spacing of 0.20 nm for the (101) plane and 0.23 nm for the (100) plane of the hexagonal close-packed (hcp) ruthenium. The images also confirmed an indication of pure crystalline nanoparticles after different chemical treatments. In addition, XRD analysis was conducted to confirm the modification of the ruthenium component with variations in pH. Appendix A shows that the crystallinity of ruthenium was maintained after the chemical treatment due to its inherent stability and inertness toward other chemical elements. Therefore, the ruthenium component on the CPPy surface maintains its composition and morphology after treatments at different pH levels.

### 3.3. Electrical Characteristics of Ru_CPPy

The electrical conductivity of Ru_CPPy was evaluated to investigate the roles of each component during charge transfers, thus characterizing the electrical properties of the nanoparticle. First, the electrical conductivity of the particles increased with an increase in the particle size of the embedded ruthenium on the carboxyl polypyrrole surface, implying that the metallic property of ruthenium nanoclusters contributed to enhancing the conductivity of the composite nanoparticles (Figure 4a). The value of electrical conductivity was also controlled by the varying proton amounts, as more acidic conditions improve the carrier density (hole) in the polymer chains (Figure 4b). We know this to be consistent because proton doping occurs during acid treatments, and β-carbon is preferentially protonated in the polymer chain structure [42]. The protonation process is critical in inducing the short-range ordering of the polymer backbone and efficient proton transport in the conjugated chains and is achieved by increasing the amount of C=C stretching [43]. Additionally, we observed a significant decline in electrical conductivity in basic conditions due to the drastic reduction in the number of charge carriers during deprotonation.

### 3.4. Hydrogen Gas Sensing Performance of the Ru_CPPy-Based Electrode

To investigate the performance of the particles as a sensor transducer, the Ru_CPPy was immobilized on the interdigitated array electrode using the spin-coating process (Appendix A). The sensing performance of the electrode was demonstrated by the catalytic effect of the ruthenium component toward hydrogen gas by a sensing mechanism [44]. Due to its reducing property demonstrated during detection, the hydrogen gas molecules that were adsorbed on the surface of ruthenium formed hydrogen atoms and provided electrons. These electrons were then transferred from the ruthenium to the polypyrrole nanoparticles, thereby reducing the charge (hole) transfer in the polymer chain. By doing so, the resistance of the sensor electrode increased. Figure 5a shows the sensing performance of the Ru_CPPy-based electrode with different ruthenium density values (Ru_CPPy_0.5, Ru_CPPy_1.5, and Ru_CPPy_3.0) toward hydrogen gas at room temperature. The minimum detectable level (MDL) was reduced to 0.5 ppm for the Ru_CPPy_3.0-based electrode. Thus, a better sensitivity to hydrogen gas was achieved with a large amount of ruthenium nanoclusters due to the effect of the enhanced catalytic activity toward hydrogen gas. To compare the role of each component in hydrogen gas sensing, measurements with only CPPyNP and Ru particles were used. The CPPyNP-based electrode exhibited no signal change due to the absence of active sites toward hydrogen gas molecules (Appendix Aa). However, the ruthenium-particle-based electrode demonstrated a reaction to high concentrations of hydrogen gas due to the small active surface area of hydrogen gas molecules (Appendix Ab). Consequently, we observed that the carboxyl polypyrrole and the ruthenium nanoclusters acted as both the charge (hole) transfer pathway and as active sites for the target analyte during sensing.

Figure 5b shows the varying responses of nanoparticles with different hydrogen concentrations as a function of ruthenium density. The Ru_CPPy_3.0-based sensor displayed nonlinear changes in response to the low concentration of hydrogen gas. However, linearity was observed over a wide range of concentrations (0.5–100 ppm). Other Ru_CPPy-based sensors with different ruthenium amounts also showed a similar tendency when they were exposed to hydrogen molecules. However, the linear behaviors of these sensors were observed in a range narrower than that of Ru_CPPy_3.0 due to the lower ruthenium densities on the surface. In addition, the Ru_CPPy-based sensor showed a more superior performance compared to other hydrogen sensors (Appendix A).

Ru_CPPy_3.0-based electrodes with different pH values were evaluated for hydrogen molecule detection to confirm the influence of proton number on the sensing performance. As shown in Figure 6a, constant ruthenium density on the surface demonstrated the same value for the MDL (0.5 ppm) of the hydrogen concentration. However, the response value of the electrodes was enhanced with increasing proton amounts (i.e., reducing pH) due to the p-type semiconductor characteristic of the polymer chain (Figure 6b). In particular, the change in response at basic conditions was greater than in other conditions due to the decline in the charge transfer pathway after the deprotonation process. Figure 6c demonstrates the changes that occurred in response to the particles as a function of the hydrogen concentration. Even though the response values varied, all of the sensor electrodes demonstrated linearity over a wide concentration range (0.5 to 100 ppm). The calculated limit of detection (LOD) in the sensor electrodes was as follows: 0.14 ppm for pH 1, 0.22 ppm for pH 4, 0.29 ppm for pH 7, 0.35 ppm for pH 10, and 0.5 ppm for pH 13. The response and recovery times of the electrodes were evaluated to further investigate the sensor’s performance (Figure 6d,e). The response and recovery times reduced from 78 to 12 s and 98 to 32 s, respectively, with an increase in the proton amounts. In other words, the electrical signals were transmitted quickly as the electrical conductivity increased. The sensing performance of the electrode was also compared with a different working temperature. According to the temperature variation, response and recovery times reduced as the temperature increased (Appendix A). In particular, recovery time decreased more rapidly than the response time due to the desorption of hydrogen gas being a faster process with the increase in working temperature.

An excellent cycle stability is essential for electrode materials in practical sensor applications. Figure 6f presents the electrical responses of the electrodes with a periodic exposure to 1 ppm of hydrogen gas at room temperature. These sensor electrodes revealed a similar response for each exposure without any changes in the response and the recovery times. As the ruthenium nanostructures were bonded on the CPPyNP surface without aggregation, they prevented the morphology disintegration of the ruthenium component during hydrogen exposure. The sensor electrodes maintained their sensing ability during the exposure to 1 ppm of hydrogen gas over 15 days (Figure 7). In addition, no morphological transitions in the nanoparticles were observed after repeated hydrogen exposures (Appendix A).

The selectivity of the sensor electrodes was also important for practical applications. Appendix A exhibits the variation in the normalized resistance of the electrode at pH 1 upon exposure to various volatile gases at low concentrations (1 ppm for H_2_ and MeOH; 100 ppm for others). Although its concentration was lower than those of other gases, H_2_ yielded a signal change that was more than five times greater than those of the other gases. Therefore, H_2_ can be distinguished from other gases based on the extent and direction of its resistance change.

## 4. Conclusions

In this study, a carboxyl polypyrrole nanoparticle decorated with ruthenium nanoclusters (Ru_CPPy) was fabricated using the facile sonochemical reduction method for applications in hydrogen sensing electrodes. The amounts and sizes of the ruthenium nanoclusters, which acted as active sites for interaction with the hydrogen molecule, were controlled by modifying the amount of RuCl_3_. The chemical state of the carboxyl polypyrrole nanoparticle was changed through the protonation process to increase the charge carrier mobility of the electrode. The as-prepared Ru_CPPy-based sensor electrode exhibited a high sensitivity (as low as 0.5 ppm) to hydrogen gas with exceptional cycle stability on account of the uniformly decorated Ru nanoclusters on the CPPy surface and the protonated polymer chain structure. Thus, in this study, we were able to demonstrate an effective approach for synthesizing composite nanomaterials with a transition metal and conducting polymer for electrochemical applications.

## Figures and Tables

**Figure 1 polymers-12-01427-f001:**
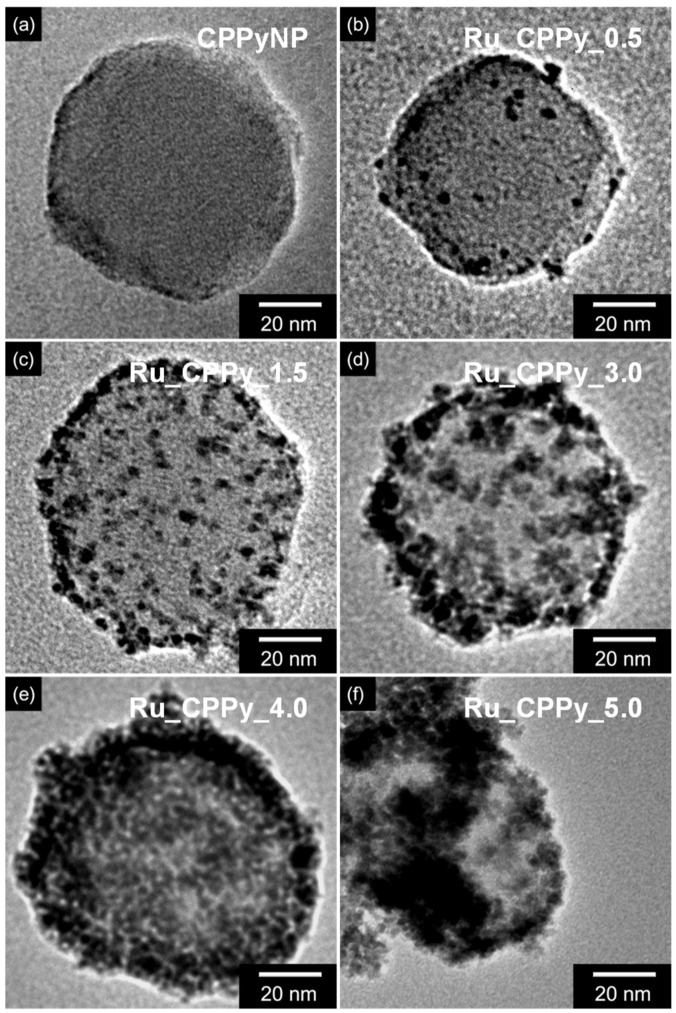
The transmission electron microscopy (TEM) images of Ru_CPPy with varying amounts of the Ru precursor (RuCl_3_): (**a**) 0 mg (CPPyNP); (**b**) 0.5 mg (Ru_CPPy_0.5); (**c**) 1.5 mg (Ru_CPPy_1.5); (**d**) 3.0 mg (Ru_CPPy_3.0); (**e**) 4.0 mg (Ru_CPPy_4.0); (**f**) 5.0 mg (Ru_CPPy_5.0).

**Figure 2 polymers-12-01427-f002:**
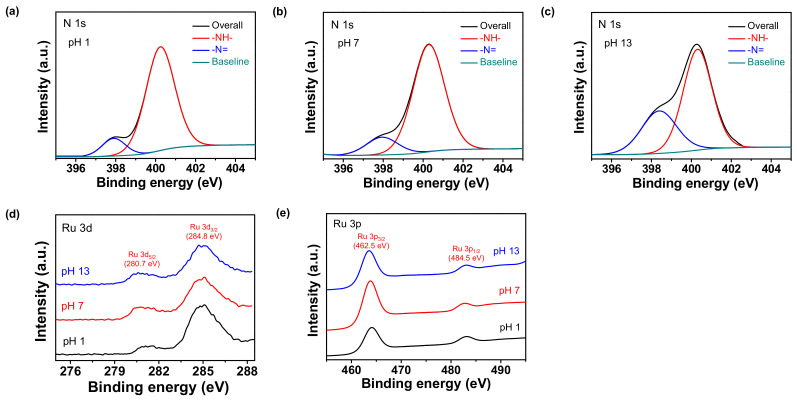
An N 1s high resolution X-ray photoelectron spectroscopy (XPS) analysis of Ru_CPPy with different pH treatments: (**a**) pH 1; (**b**) pH 7; (**c**) pH 13. (**d**) Ru 3d and (**e**) Ru 3p XPS with diverse pH states (black: pH 1; red: pH 7; blue: pH 13).

**Figure 3 polymers-12-01427-f003:**
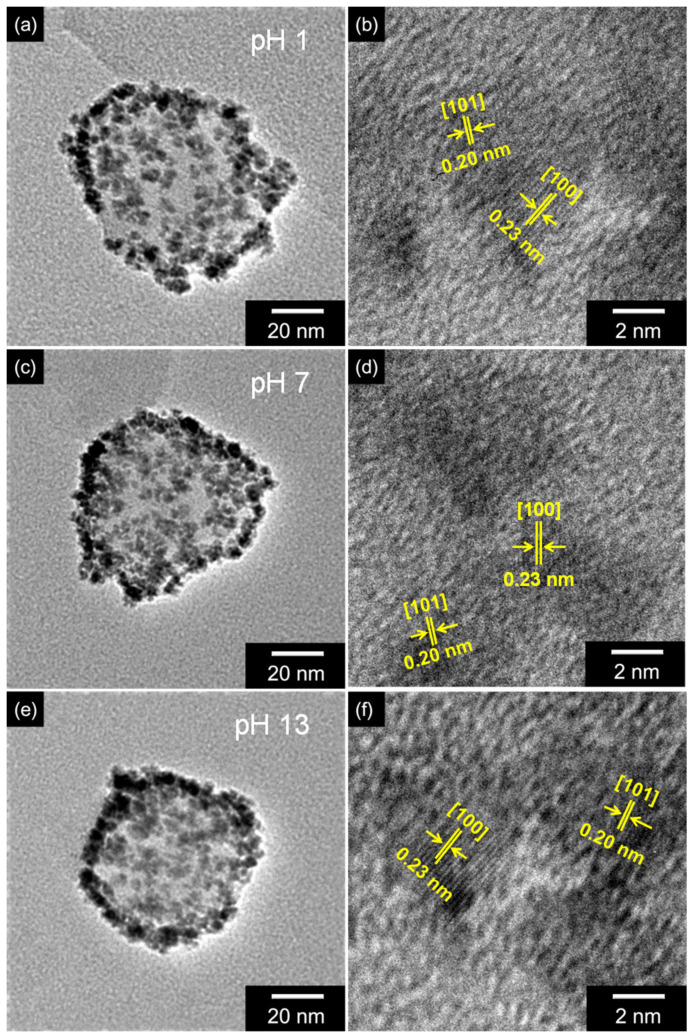
TEM and high-resolution transmission electron microscopy (HR-TEM) images of Ru_CPPy with different pH values: (**a**,**b**) pH 1; (**c**,**d**) pH 7; (**e**,**f**) pH 13.

**Figure 4 polymers-12-01427-f004:**
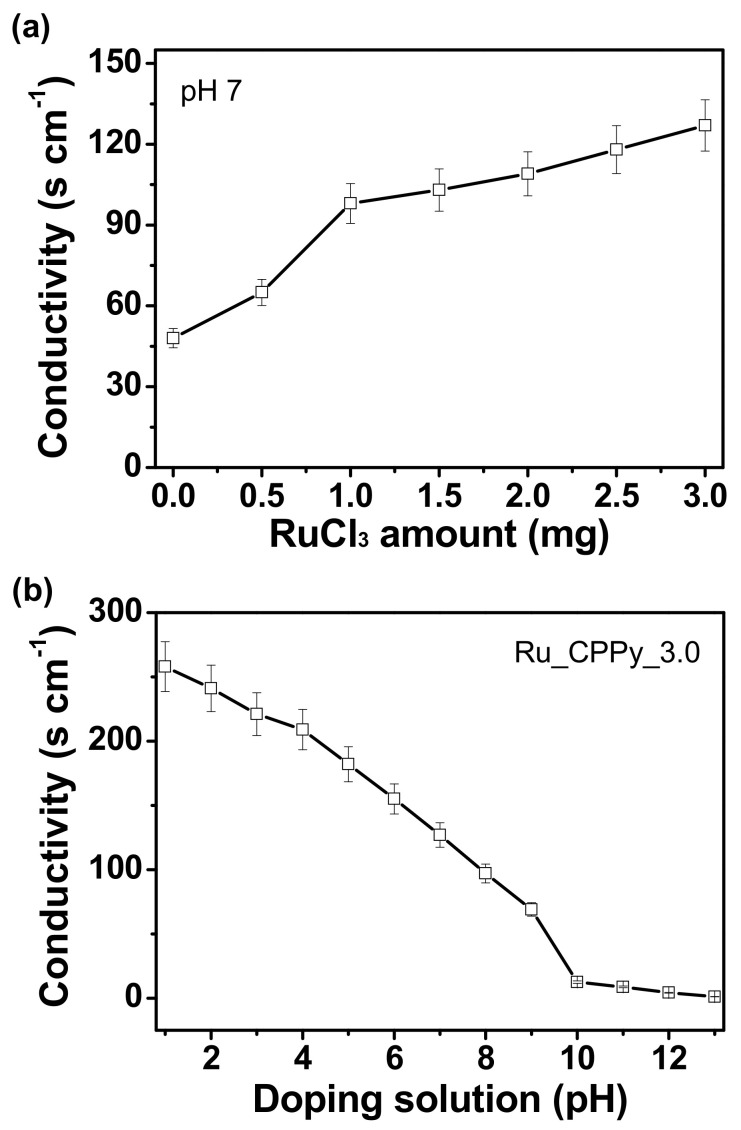
The electrical conductivities of (**a**) different CPPyNP-based nanoparticles with diverse RuCl_3_ concentrations with a pH of 7 and (**b**) Ru_CPPy_3.0 with various pH values in a chemical treatment.

**Figure 5 polymers-12-01427-f005:**
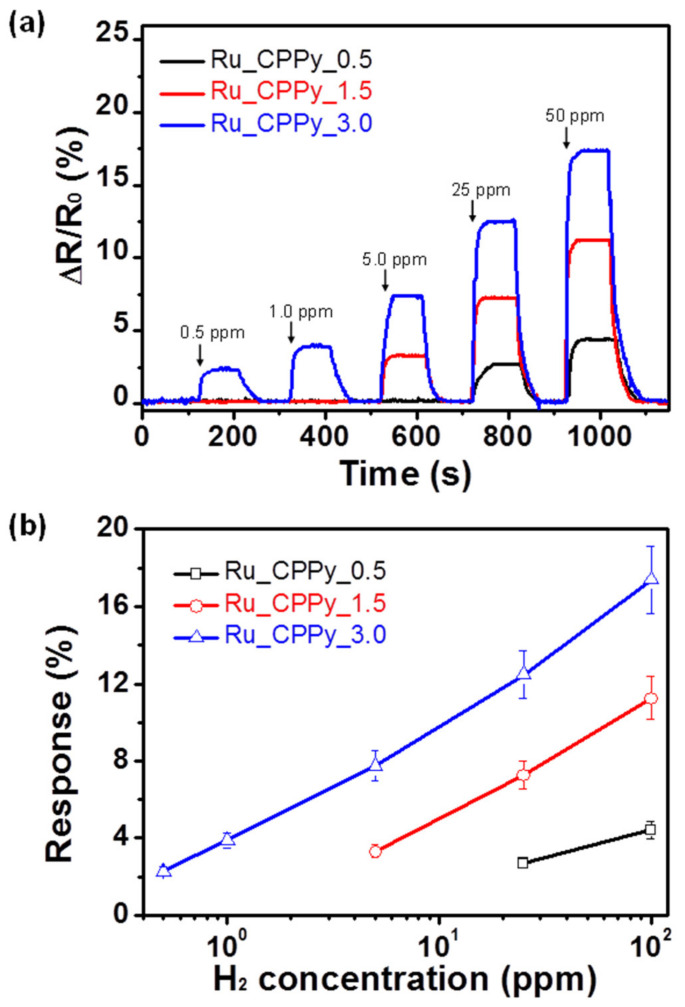
(**a**) The normalized resistance changes upon sequential exposure to various concentrations of hydrogen gas. (**b**) Calibration curves serve as a function of the hydrogen gas concentrations (black: Ru_CPPy_0.5; red: Ru_CPPy_1.5; blue: Ru_CPPy_3.0).

**Figure 6 polymers-12-01427-f006:**
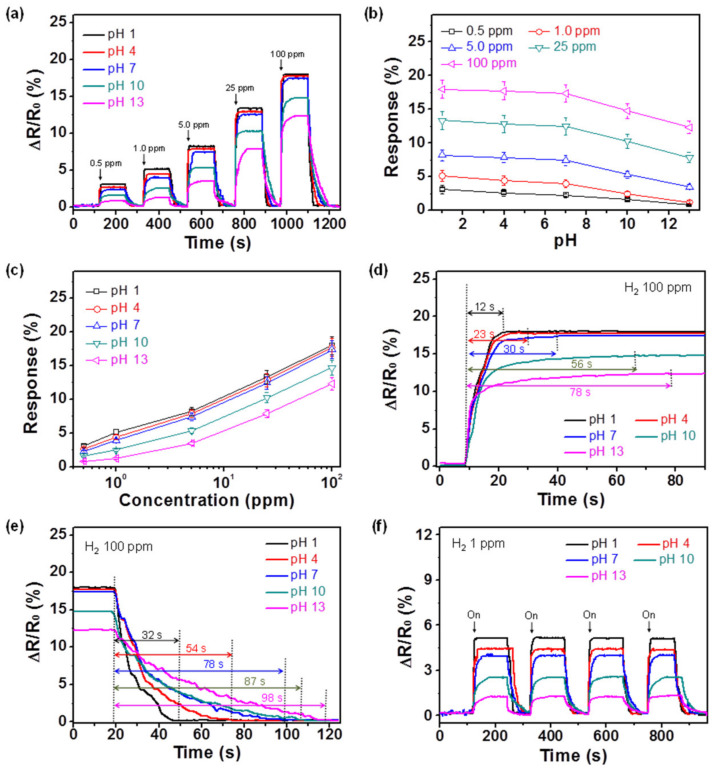
(**a**) The normalized resistance changes upon sequential exposure to various concentrations of hydrogen gas (black: pH 1; red: pH 4; blue: pH 7; green: pH 10; pink: pH 13). (**b**) The response changes upon sequential pH differences with same hydrogen concentration (black: 0.5 ppm; red: 1 ppm; blue: 5 ppm; green: 25 ppm; pink: 100 ppm). (**c**) Calibration lines serve as a function of the hydrogen gas concentration. (**d**) The response and (**e**) recovery times of the electrodes with proton variations at 100 ppm of hydrogen gas. (**f**) The normalized resistance changes of the electrodes upon sequential periodic exposure to 1 ppm of hydrogen gas. Each pH treatment is represented by the following: black for pH 1; red for pH 4; blue for pH 7; green for pH 10; and pink for pH 13, respectively.

**Figure 7 polymers-12-01427-f007:**
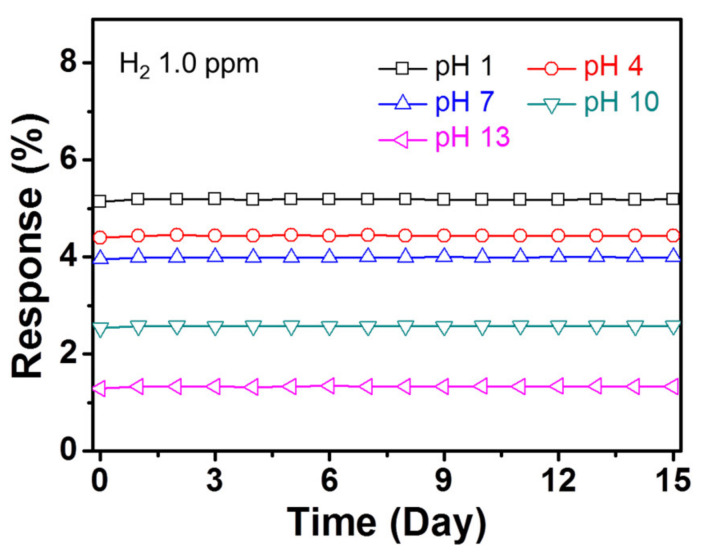
The normalized resistance changes of the electrodes with periodic exposures to 1 ppm of hydrogen gas for 15 days (black: pH 1; red: pH 4; blue: pH 7; green: pH 10; pink: pH 13).

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
