# Peer review of "Ruthenium Decorated Polypyrrole Nanoparticles for Highly Sensitive Hydrogen Gas Sensors Using Component Ratio and Protonation Control"

_polymers, 2020, doi:10.3390/polym12061427_

Round 1

Reviewer 1 Report

This paper studied a nanocomposite comprising ruthenium nanoclusters decorated on carboxyl polypyrrole nanoparticles (Ru_CPPy) to detect hydrogen gas through a simple sonochemistry method. The size and population of the ruthenium nanoclusters were optimized by controlling the amount of ruthenium precursor.

The review comments for this paper are shown below:

Line 27, the authors introduced the importance of the hydrogen gas. However, some contents are repeated in the text such as using hydrogen for fuel production. Please reduce the introduction of the hydrogen importance.

Line 36, the authors claimed that "Hydrogen gas sensors using various sensing materials have been widely studied in many research groups for more than two decades". -- What is the status of the usage of hydrogen sensor in industries?

In Introduction, the authors need to mention and introduce some works detailedly from the literature.

For the composite materials, please add some recent works into the Introduction (10.3390/nano9101397; 10.1166/jnn.2019.16584; etc.) and show the developments of this field. The differences of inorganic and organic materials should be discussed. 

In Section 2.1, please provide the city and country information of the chemicals and materials.

In Section 2.5, please separate all the characterization methods and introduce them more clearly.

In Section 3.1, the effect of the nanoparticles in the solution should be discussed.

In Figure 3, how could you measure the length of (101) plane and (100) plane in the figures?

In Figure 4b, why did the conductivity not change much when the pH of the doping solution is 10?

In Section 3.4, what is the difference between your hydrogen sensor and the previous work in terms of sensing time? How could you improve it?

What is the possibility to use the nanomaterials to detect other gases?

Please double check the English writing before the resubmission.

Author Response

COMMENT

This paper studied a nanocomposite comprising ruthenium nanoclusters decorated on carboxyl polypyrrole nanoparticles (Ru_CPPy) to detect hydrogen gas through a simple sonochemistry method. The size and population of the ruthenium nanoclusters were optimized by controlling the amount of ruthenium precursor.

The review comments for this paper are shown below:

REPONSE:

We highly appreciate your comments on our manuscript.

COMMENT 1) Line 27, the authors introduced the importance of the hydrogen gas. However, some contents are repeated in the text such as using hydrogen for fuel production. Please reduce the introduction of the hydrogen importance.

REPONSE:

Thank for your comment about introduction of the manuscript. As you commented, we modified line 27 in the revised manuscript.

Revised parts in the manuscript

(1) The sentence “Hydrogen is one of the most popular raw materials used in diverse industrial processes, such as fuel production, chemical synthesis, and plant operation, owing to its important advantages, such as its high availability in nature, high conversion efficiency, non-toxicity, and feasibility of development from renewable sources [1-3].” was changed to “Hydrogen is one of the most popular raw materials used in diverse industrial processes owing to its important advantages, such as its high availability in nature, high conversion efficiency, non-toxicity, and feasibility of development from renewable sources [1,2].” in the revised manuscript. (see Line 27, Page 1)

COMMENT 2) Line 36, the authors claimed that "Hydrogen gas sensors using various sensing materials have been widely studied in many research groups for more than two decades". -- What is the status of the usage of hydrogen sensor in industries?

REPONSE:

Thank you for your valuable comment of the introduction. As you mentioned, we added explain about current status of hydrogen gas application.

Revised parts in the manuscript

(1) The sentence “Recently, palladium nanomaterial-based sensors have been developed that can detect down to ppm level at room temperature.[9-11]” was added in the revised manuscript. (see Line 36, Page 1)

COMMENT 3) In Introduction, the authors need to mention and introduce some works detailedly from the literature.

REPONSE:

Thank you for your comment about illustration of some works in hydrogen sensor application. As you mentioned, we have described the recent hydrogen sensor study as an example.[R1]

[R1] ACS Appl. Mater. Interfaces 2019, 11, 13343-13349.

Revised parts in the manuscript

(1) The sentence “Cho et al. synthesized half-pipe palladium nanotube-based networks to apply them into the flexible hydrogen gas sensor.[11]” in the revised manuscript. (see Line 37, Page 1)

COMMENT 4) For the composite materials, please add some recent works into the Introduction (10.3390/nano9101397; 10.1166/jnn.2019.16584; etc.) and show the developments of this field. The differences of inorganic and organic materials should be discussed.

REPONSE:

We added some recent works about composite materials that are suggested by reviewer 1.[R2,R3] We also added illustration about differences of inorganic and organic components in the composite.

[R2] Nanomaterials 2019, 9, 1397.

[R3] J. Nanosci. Nanotechnol. 2019, 19, 3173.

Revised parts in the manuscript

(1) [R2 and R3] were added as [reference 19 and 20] in the revised manuscript.

(2) The sentence “As a sensor electrode material, the organic component (i.e. sp2 carbon) acts as a charge transfer pathway and template, whereas inorganic component operates as binding site to target analyte.” was inserted in the revised manuscript. (see Line 47, Page 2)

COMMENT 5) In Section 2.1, please provide the city and country information of the chemicals and materials.

REPONSE:

Thank you for your comment in the experimental section. As you mentioned, we added city and country information of the chemicals and materials in the revised manuscript.

COMMENT 6) In Section 2.5, please separate all the characterization methods and introduce them more clearly.

REPONSE:

Thank you for your valuable comment about experimental section of the manuscript. As you mentioned, we modified section 2.5 in the revised manuscript.

Revised parts in the manuscript

(1) The sentence “High resolution transmission electron microscopy (HR-TEM) and field-emission scanning electron microscopy images were collected using JEOL 6700 and JEOL JEM-200CX (JEOL Ltd., Tokyo, Japan), respectively.” was changed to “Field-emission scanning electron microscopy (FE-SEM) images were collected using JEOL JEM-200CX (JEOL Ltd., Tokyo, Japan).” in the revised manuscript. (see Line 110, Page 3)

COMMENT 7) In Section 3.1, the effect of the nanoparticles in the solution should be discussed.

REPONSE:

We appreciate your comment about effect of the nanoparticles in the solution. As you mentioned, we conducted additional fabrication of Ru particles without CPPyNPs to confirm role of CPPy during chemical reaction. As shown in Figure R1, a large Ru particle (ca. 800 nm) was generated with a clump. Thus, CPPyNP is able to act as a framework for creating uniform Ru nanoparticles.

Figure R1. TEM image of Ru particles without CPPyNPs.

Revised parts in the manuscript

(1) Figure R1 was added as Figure S3 in the revised manuscript.

(2) The sentences “In order to check the role of CPPy in Ru particle generation, the reaction was conducted except CPPyNP, and a large Ru particle (ca. 800 nm) was generated with a clump (Figure S3). Thus, CPPyNP is able to act as a framework for creating uniform Ru nanoparticles.” were added in the revised manuscript. (See Line 134, Page 3)

COMMENT 8) In Figure 3, how could you measure the length of (101) plane and (100) plane in the figures?

REPONSE:

Thank you for your comment about measuring method of d-spacing of the Ru component. We used fast fourier transformation (FTT) of the HR-TEM to measure d-spacing of the Ru particles.

COMMENT 9) In Figure 4b, why did the conductivity not change much when the pH of the doping solution is 10?

REPONSE:

Thank you for your comment about conductivity of the particles. As pH increases, the number of hole in the polymer chain decreases and the electrical conductivity also decreases. This phenomenon progresses further in the base solution, resulting in a significant reduction in electrical conductivity. In particular, in conditions above pH 10, the numerical reduction of hole has been sufficiently achieved, so it has almost a similar electrical conductivity value.

COMMENT 10) In Section 3.4, what is the difference between your hydrogen sensor and the previous work in terms of sensing time? How could you improve it?

REPONSE:

Thank you for your comment about sensing time of the electrode. We introduced Pd particles into polypyrrole conductive film and used it as hydrogen sensor electrode.[R4] However, in this case, despite the introduction of a large amount of Pd, it was not quickly sensed due to uneven size and clumping. Thus, detection time and recovery time were more than 100 seconds and 300 seconds, respectively.

On the other hand, in this experiment, small-sized Ru particles were uniformly distributed to form a large active surface area, showing fast reaction times. This result is due to the uniform distribution of Ru nanoparticles on the CPPy surface.

[R4] J. Mater. Sci. 55, 5156 (2020)

COMMENT 11) What is the possibility to use the nanomaterials to detect other gases?

REPONSE:

Thank you for your mention about selectivity of the sensor electrode. As you commented we measured sensing performance of the electrode to other chemicals. Figure R2 exhibits the variation in the normalized resistance of the electrode at pH 1 upon exposure to various volatile gases at low concentrations (1 ppm for H2 and MeOH; 100 ppm for others). Although its concentration is lower than those of other gases, H2 yields a signal change that is more than five times larger compared to those of the other gases. Therefore, H2 can be distinguished from other gases based on the extent and direction of the resistance change.

Figure R2. Normalized resistance changes of the electrode do different analytes.

Revised parts in the manuscript

(1) The Figure R2 was added as Figure S12 in the revised supporting information.

(2) The paragraph “The selectivity of the sensor electrodes is also important for practical applications. Figure S12 exhibits the variation in the normalized resistance of the electrode at pH 1 upon exposure to various volatile gases at low concentrations (1 ppm for H2 and MeOH; 100 ppm for others). Although its concentration is lower than those of other gases, H2 yields a signal change that is more than five times larger compared to those of the other gases. Therefore, H2 can be distinguished from other gases based on the extent and direction of the resistance change.” was inserted in the revised manuscript. (see Line 288, Page 11)

COMMENT 12) Please double check the English writing before the resubmission.

REPONSE:

As you mentioned, we modified grammatically errors in the revised manuscript.

----------------------------------------------------------------------------------------------------------

The revised parts were highlighted by yellow background color. Thank you for your critical and valuable comments again.

Sincerely,

Prof. Junseop Lee

Reviewer 2 Report

The submitted manuscript presents hydrogen detection using a nanocomposite comprising ruthenium nanoclusters decorated on carboxyl polypyrrole nanoparticles. The interesting results are shown in the paper. However, some points have to be clarified or improved. Therefore, this manuscript could be consider for publication after major revision. The Authors are kindly requested to consider following remarks:

  1. Did the Authors investigate selectivity of the sensor? The Authors should provide information, whether they measured the sensor responses to other common gasses (e.g. water vapor (humidity), oxygen, carbon dioxide).
  2. The sensing properties of Ru_CPPy were examined in the hydrogen concentration range from 0.5 ppm to 100 ppm. The value of 0.5 ppm is the lowest concentration investigated. It does not mean a detection limit of the sensor. Please, try to calculate hydrogen detection limit using definitions known in the literature.
  3. The Authors are requested to consider mention in their article other papers describing similar procedure for gas sensors fabrication, i.e. sonochemical synthesis of nanomaterial and its deposition on the array of microelectrodes (e.g. Talanta 189 (2018) 225, Nanoscale Research Letters (2017) 12:97, Ultrasonics 69 (2016) 67).
  4. The electrical responses were measured only at room temperature. What is an influence of the operating temperature on sensor response?
  5. The sensing mechanism of Ru_CPPy sensor is completely omitted in the paper. Is the investigated nanomaterial p-type or n-type semiconductor? What is the nature of the interaction between hydrogen molecules and Ru_CPPy film (physical or chemical)? The Authors have to propose sensing mechanism.

Author Response

COMMENT

The submitted manuscript presents hydrogen detection using a nanocomposite comprising ruthenium nanoclusters decorated on carboxyl polypyrrole nanoparticles. The interesting results are shown in the paper. However, some points have to be clarified or improved. Therefore, this manuscript could be consider for publication after major revision. The Authors are kindly requested to consider following remarks:

REPONSE:

We highly appreciate your comments on our manuscript.

COMMENT 1) Did the Authors investigate selectivity of the sensor? The Authors should provide information, whether they measured the sensor responses to other common gasses (e.g. water vapor (humidity), oxygen, carbon dioxide).

REPONSE:

Thank you for your mention about selectivity of the sensor electrode. As you commented we measured sensing performance of the electrode to other chemicals. Figure R3 exhibits the variation in the normalized resistance of the electrode at pH 1 upon exposure to various volatile gases at low concentrations (1 ppm for H2 and MeOH; 100 ppm for others). Although its concentration is lower than those of other gases, H2 yields a signal change that is more than five times larger compared to those of the other gases. Therefore, H2 can be distinguished from other gases based on the extent and direction of the resistance change.

Figure R3. Normalized resistance changes of the electrode do different analytes.

Revised parts in the manuscript

(1) The Figure R3 was added as Figure S12 in the revised supporting information.

(2) The paragraph “The selectivity of the sensor electrodes is also important for practical applications. Figure S12 exhibits the variation in the normalized resistance of the electrode at pH 1 upon exposure to various volatile gases at low concentrations (1 ppm for H2 and MeOH; 100 ppm for others). Although its concentration is lower than those of other gases, H2 yields a signal change that is more than five times larger compared to those of the other gases. Therefore, H2 can be distinguished from other gases based on the extent and direction of the resistance change.” was inserted in the revised manuscript. (see Line 288, Page 11)

COMMENT 2) The sensing properties of Ru_CPPy were examined in the hydrogen concentration range from 0.5 ppm to 100 ppm. The value of 0.5 ppm is the lowest concentration investigated. It does not mean a detection limit of the sensor. Please, try to calculate hydrogen detection limit using definitions known in the literature.

REPONSE:

Thank you your valuable comment to about calculated limit of detection (LOD) in the manuscript. As you commented, we calculated limit of detection (LOD) of the IDA sensor electrode using theoretical equations.[R5-R7] As a result, the values of LOD are show following as: 0.14 ppm for pH 1; 0.22 ppm for pH 4; 0.29 ppm for pH 7; 0.35 ppm for pH 10; 0.5 ppm for pH 13.

[R5] ACS Appl. Mater. Interfaces, 2019, 11, 2364-2373.

[R6] Adv. Funct. Mater. 2016, 26, 7462-7469.

[R7] J. Mater. Chem. A 2018, 6, 478-488.

Revised parts in the manuscript

(1) The sentence “Additionally, calculated limit of detection (LOD) of the sensor electrodes is suggested following as: 0.14 ppm for pH 1; 0.22 ppm for pH 4; 0.29 ppm for pH 7; 0.35 ppm for pH 10; 0.5 ppm for pH 13.” was added in the revised manuscript. (see Line 270, Page 10)

COMMENT 3) The Authors are requested to consider mention in their article other papers describing similar procedure for gas sensors fabrication, i.e. sonochemical synthesis of nanomaterial and its deposition on the array of microelectrodes (e.g. Talanta 189 (2018) 225, Nanoscale Research Letters (2017) 12:97, Ultrasonics 69 (2016) 67).

REPONSE:

We cited research papers that are suggested from reviewer 2.[R8-R10] We also added explanation about sonochemical synthesis of nanomaterial and its deposition on the array of microelectrodes in the introduction part.

[R8] Talanta 2018, 189, 225-232.

[R9] Nanoscale Res. Lett. 2017, 12, 97.

[R10] Ultrasonics 2016, 69, 67-73.

Revised parts in the manuscript

(1) [R8-R10] were inserted as [reference 21-23] in the revised manuscript.

(2) The sentence “In addition, recently, a new synthetic process of sensor electrode materials has been developed using facile sonochemical method.[21-23]” was added in the revised manuscript. (see Line 47, Page 2)

COMMENT 4) The electrical responses were measured only at room temperature. What is an influence of the operating temperature on sensor response?

REPONSE:

Thank you for your valuable comment about sensing performance with temperature variations. According to the sensing performance changes of the sensor electrode with temperature, response and recovery time were reduced as the temperature increased. In particular, recovery time decreases more rapidly than that of response time owing to desorption of hydrogen gas is faster with increasing working temperature.

Figure R4. Response and recovery time changes of the pH 1 electrode with working temperature variation.

Revised parts in the manuscript

(1) Figure R4 was added as Figure S10 in the revised supporting information.

(2) The sentences “Sensing performance of the electrode is also compared with different working temperature. According to the temperature variation, response and recovery time were reduced as the temperature increased (Figure S10). In particular, recovery time decreases more rapidly than that of response time owing to desorption of hydrogen gas is faster with increasing working temperature.” were inserted in the revised manuscript. (see Line 274, Page 10)

COMMENT 5) The sensing mechanism of Ru_CPPy sensor is completely omitted in the paper. Is the investigated nanomaterial p-type or n-type semiconductor? What is the nature of the interaction between hydrogen molecules and Ru_CPPy film (physical or chemical)? The Authors have to propose sensing mechanism.

REPONSE:

Thank you for your comment about sensing mechanism of the electrode. As you mentioned, we added sensing mechanism in the revised manuscript.

Revised parts in the manuscript

(1) The sentences “The sensing performance of the electrode is demonstrated by the catalytic effect of the ruthenium component toward hydrogen gas as following sensing mechanism [44]. Owing to the reducing property demonstrated during detection, the hydrogen gas molecules adsorbed on the surface of ruthenium formed hydrogen atoms and provided electrons. These provided electrons were transferred from ruthenium to polypyrrole nanoparticles, thereby reducing the charge (hole) transfer in the polymer chain. Therefore, the resistance of the sensor electrode was increased.” (see Line 233, Page 10)

----------------------------------------------------------------------------------------------------------

The revised parts were highlighted by yellow background color. Thank you for your critical and valuable comments again.

Sincerely,

Prof. Junseop Lee

Reviewer 3 Report

The hydrogen gas sensor based on Ruthenium decorated polypyrrole is researched in detail. The exploration is profound for the development of hydrogen sensors, because the material is original and the experiment is scientific and reasonable. Therefore, I think this work is valuable to be published in “Polymers” after a revised.

  1. There are many English grammar and usage mistakes in this work. Please revise it carefully again
  2. Please explain the “active surface area of the hydrogen molecules (line 19)”.
  3. The definitions of “Protonation Control”, “protonation level” and “protonation states” should be provided.
  4. More acid conditions means a higher protonation level, which improve the carrier density (hole) in the polymer chains. Why it is helpful to improve the hydrogen sensing performance? More explain should be provided.
  5. More comparison with the other types of hydrogen sensors is necessary to express the advantages and prospects about the novel H2 sensor. Such as J. Tian, et al., International Journal of Hydrogen Energy 2020, 45, 14594.; L. Zhang, et al., Nanotechnology 2019, 31, 015504.

Author Response

COMMENT

The hydrogen gas sensor based on Ruthenium decorated polypyrrole is researched in detail. The exploration is profound for the development of hydrogen sensors, because the material is original and the experiment is scientific and reasonable. Therefore, I think this work is valuable to be published in “Polymers” after a revised.

REPONSE:

We highly appreciate your comments on our manuscript.

COMMENT 1) There are many English grammar and usage mistakes in this work. Please revise it carefully again

REPONSE:

Thank you for your valuable comment about grammatical errors in the manuscript. As you mentioned, we double checked it.

COMMENT 2) Please explain the “active surface area of the hydrogen molecules (line 19)”.

REPONSE:

Thank you for your comment about the ambiguous expression in the abstract part. We originally tried to explain that Ru nanoparticles have a high active surface area for H2 molecule. In order to clarify the meaning of the sentence, we modified “active surface area of the hydrogen molecules” to “active surface area to the target analyte (hydrogen molecule)”.

Revised parts in the manuscript

(1) The sentence “The morphology and density control of the ruthenium component increases the active surface area of the hydrogen molecules.” was changed to “The morphology and density control of the ruthenium component increases the active surface area to the target analyte (hydrogen molecule).” in the revised manuscript. (see Line 18, Page 1)

COMMENT 3) The definitions of “Protonation Control”, “protonation level” and “protonation states” should be provided.

REPONSE:

Thank you for your valuable comments about terminology in the manuscript. The definition of “protonation control” is modifying amount of proton in the polymer chain by different pH aqueous solutions. Then, we changed the words “protonation level” and “protonation states” to “proton amount” in the revised manuscript. The “proton amount” is controlled using different pH value of solutions.

COMMENT 4) More acid conditions means a higher protonation level, which improve the carrier density (hole) in the polymer chains. Why it is helpful to improve the hydrogen sensing performance? More explain should be provided.

REPONSE:

Thank you for your comments about effect of protonation control on the sensing performance. Adjusting proton amount in the polymer chain does not increase the active surface area to hydrogen gas. However, we have confirmed that the response time and recovery time are reduced by the reduce pH of solution. This is because electrical signals are transmitted quickly as electrical conductivity increases.

Revised parts in the manuscript

(1) The sentence “In other words, electrical signals are transmitted quickly as electrical conductivity increases.” was added in the revised manuscript. (see Line 274, Page 10)

COMMENT 5) More comparison with the other types of hydrogen sensors is necessary to express the advantages and prospects about the novel H2 sensor. Such as J. Tian, et al., International Journal of Hydrogen Energy 2020, 45, 14594.; L. Zhang, et al., Nanotechnology 2019, 31, 015504.

REPONSE:

Thank you for your comment about comparison of other hydrogen sensors. As you mentioned, we added more research papers that are suggested by reviewer 3.[R11,R12]

[R11] Int. J. Hydrog. Energy 2020, 45, 14594-14601.

[R12] Nanotechnology 2020, 31, 015504.

Revised parts in the manuscript

(1) The [R11 and R12] were added as [reference 9 and 10] in the revised manuscript.

----------------------------------------------------------------------------------------------------------

The revised parts were highlighted by yellow background color. Thank you for your critical and valuable comments again.

Sincerely,

Prof. Junseop Lee

Round 2

Reviewer 1 Report

The authors have improved the paper after the revisions.

Reviewer 2 Report

The manuscript entitled “Ruthenium Decorated Polypyrrole Nanoparticles for Highly Sensitive Hydrogen Gas Sensors Using Component Ratio and Protonation Control” has been significantly improved. Authors answered all questions and revised manuscript by inserting the new results into the paper. Authors investigated selectivity of the sensor, determined its detection limit, extended list of the references, studied an influence of the operating temperature on sensor response, and proposed sensing mechanism. Therefore, I recommend this manuscript for publication.

Reviewer 3 Report

The manuscript has been greatly improved, I think it can be published now.